# Spectroscopic Discrimination of Bee Pollen by Composition, Color, and Botanical Origin

**DOI:** 10.3390/foods10081682

**Published:** 2021-07-21

**Authors:** Roman Bleha, Tetiana V. Shevtsova, Martina Živčáková, Anna Korbářová, Martina Ježková, Ivan Saloň, Ján Brindza, Andriy Synytsya

**Affiliations:** 1Department of Carbohydrates and Cereals, University of Chemistry and Technology Prague, Technická 5, 166 28 Prague, Czech Republic; shevtsovat@ukr.net (T.V.S.); andrej.sinica@vscht.cz (A.S.); 2Department of Physics and Measurements, University of Chemistry and Technology Prague, Technická 5, 166 28 Prague, Czech Republic; tina.zivcakova@gmail.com (M.Ž.); anna.korbarova@workswell.cz (A.K.); 3Department of Chemical Engineering, University of Chemistry and Technology Prague, Technická 5, 166 28 Prague, Czech Republic; jezkovar@vscht.cz (M.J.); saloni@vscht.cz (I.S.); 4Institute of Biological Conservation and Biosafety, Slovak University of Agriculture in Nitra, Tr. A. Hlinku 2, 949 76 Nitra, Slovakia; Jan.Brindza@uniag.sk

**Keywords:** bee pollen, discrimination, spectroscopic methods, image analysis, color

## Abstract

Bee pollen samples were discriminated using vibrational spectroscopic methods by connecting with botanical sources, composition, and color. SEM and light microscope images of bee pollen loads were obtained and used to assess the botanical origin. Fourier transform (FT) mid- and near-infrared (FT-MIR, FT-NIR), and FT-Raman spectra of bee pollen samples (a set of randomly chosen loads can be defined as an independent sample) were measured and processed by principal component analysis (PCA). The CIE L*a*b* color space parameters were calculated from the image analysis. FT-MIR, FT-NIR, and FT-Raman spectra showed marked sensitivity to bee pollen composition. In addition, FT-Raman spectra indicated plant pigments as chemical markers of botanical origin. Furthermore, the fractionation of bee pollen was also performed, and composition of the fractions was characterized as well. The combination of imaging, spectroscopic, and statistical methods is a potent tool for bee pollen discrimination and thus may evaluate the quality and composition of this bee-keeping product.

## 1. Introduction

Bee pollen is a mixture of pollen grains, nectar, and bee excreta. This apiculture product is used in food and cosmetics due to the high content of many nutrients and bioactive compounds including amino acids, lipids, sugars, and pigments [1,2,3,4]. Bee pollen varies significantly according to the chemical and botanical composition that leads to the necessity of its effective identification and sorting [5].

Bee pollen may have unifloral or polyfloral origins. Unifloral bee pollen loads are collected by a labor bee from flowers of a single botanical species. In this case, pollen grains have the same morphology and chemical composition; as such bee pollen could easily be assigned to one of the plant species by microscopic identification of its pollen grains. By contrast, if the source of pollen is restricted, labor bees visit the flowers of different plant species. As a result, different pollen grains are mixed in the loads yielding polyfloral bee pollen. Such pollen loads may show variable properties depending on the ratio between botanical species. Evaluation of such materials is complicated [6,7].

Color of pollen grains depends on botanical species and mature degree. Two groups of structurally different plant pigments were found in flower pollen: carotenoids, hydrocarbons with conjugated CC bonds, and flavonoids, aromatics containing hydroxylic and other polar groups. Bee pollen color is determined by the composition of pollen grain pigments. Bee pollen having pollen grains of the same color is monochromatic, otherwise it is polychromatic. There is no direct connection between color and botanical origin [3], so mono/polychromatic bee pollen could be unifloral or polyfloral. Nevertheless, visual color estimation has been widely used for the identification of bee pollen.

Bee pollen classification is necessary for the estimation of their nutritional value, authenticity, and safety. There are many analytical methods used for the evaluation of bee pollen quality [2], but they are often time and work consuming and are restricted to a limited number of analytes. Moreover, some of them need chemical modification that leads to a loss of unstable compounds. The solution could be the use of rapid nondestructive screening methods and approaches based on microscopic and/or spectroscopic techniques [8]. Scanning electron microscopy (SEM) can identify pollen grains directly in the bee pollen granule and detect pollen admixtures as well [9,10]. Light microscopy can be used for image and color analysis of bee pollen loads; pollen grains can be identified in the extracts [11]. Vibration spectroscopic methods are sensitive to the chemical composition of bee pollen and pollen grains [8,12,13,14,15,16,17]. In addition, statistical evaluation of the image spectra may be useful for the evaluation of bee pollen homogeneity and for the discrimination of bee pollen samples yielding trends and clusters [8,16]. In our previous works [8,9,18,19], vibrational spectroscopy and color analysis based on diffuse reflectance VIS spectra were used in the analysis of homogenates of bee pollen and bee bread; however, this approach is not able to evaluate the botanical and color heterogeneity of these beekeeping products.

Current work is devoted to the spectroscopic identification and discrimination of bee pollen according to the difference in composition and color connected with the botanical source. Our new approach is based on the conception that each randomly selected load of bee pollen can be defined as an independent sample.

## 2. Materials and Methods

### 2.1. Sampling of Bee Pollen

Eleven bee pollen samples used for analysis (Table 1) were collected from various localities and regions of the Slovak Republic. Reference samples of sunflower (*Helianthus annui*) and goat willow (*Salix caprea*) pollen grains originated from Kralupy nad Vltavou and Roztoky u Prahy (Central Bohemian region, Czech Republic), respectively. The samples were stored in a freezer at −20 °C. The botanical origin of bee pollens has been established by beekeepers and should be confirmed by analyses. These samples come from a variety of botanical species; it is expected that some of them are unifloral and monochromatic, but some are polyfloral and/or polychromatic. One hundred pollen loads of each sample were randomly selected for recording of the FT-MIR and FT-NIR spectra using attenuated total reflectance (ATR) and diffuse reflectance (DR) techniques, respectively. Similarly, ten pollen loads of each sample were randomly selected for recording of the FT-Raman spectra. The lower number of bee pollen loads is explained by the greater laboriousness of Raman measurements compared to FT-IR.

### 2.2. Spectroscopic Measurements

FT-MIR ATR spectra (range 650–4000 cm^−1^, resolution 2 cm^−1^, 64 scans) were recorded on FT-IR spectrometer Nicolet 6700 (Thermo Fisher Scientific, Waltham, MA, USA) using KBr beam splitter and smart MIRacle holder. FT-NIR DR spectra (range 10,000–4000 cm^−1^, resolution 8 cm^−1^, 64 scans) were recorded on the same spectrometer using CaF_2_ beam splitter and a smart NIR UpDRIFT holder. FT-Raman spectra (range 100–4000 cm^−1^, 1000 scans, resolution 4 cm^−1^) were recorded with FT-Raman spectra were measured with Nicolet iS50 with FT Raman module (Thermo Fisher Scientific, Waltham, MA, USA), Nd:YAG laser (λ_ex_ = 1064 nm, power 500 mW), CaF_2_ beam splitter and InGaAs detector. Vibration spectra were recorded and processed using software Omnic 8.0 and 9.0 (Thermo Fisher Scientific, Waltham, MA, USA). The raw spectra were exported in ASCII format to Origin 6.0 software (Microcal Origin, Northampton, MA, USA) for further processing (smoothing, baseline correction), calculation of the average spectra of each sample recorded for 100 (FT-MIR, FT-NIR) or 10 (FT-Raman) randomly chosen pollen loads and preparation of the graphs.

### 2.3. Principal Component Analysis of the Spectra

The FT-MIR, FT-NIR, and FT-Raman data were imported to The Unscrambler X 10.1 (Aspen Technology Inc., Bedford, MA, USA) and/or TQ Analyst (Thermo Fisher Scientific, Waltham, MA, USA) software for statistical evaluation. Principal component analysis (PCA) was used for discrimination of bee pollen samples based on their spectroscopic characteristics.

### 2.4. Image and Color Analysis

SEM images of bee pollen loads were obtained using electron microscopes ZEISS EVO LS 15 (Carl Zeiss, Germany) and TESCAN VEGA3 LMU (TESCAN ORSAY HOLDING, Czech Republic). Image and color analysis of bee pollen loads was made using a stereomicroscope STEMI 2000 (Carl Zeiss AG, Oberkochen, Germany) with LED SCHOTT MC 1500 and a digital camera Sony DFW-SX 910 (Sony, Minato, Tokyo, Japan). The CIE L*a*b* color space parameters of bee pollen loads were obtained from light microscope images using NIS-Elements 2.30 software (Laboratory Imaging s.r.o., Prague, Czech Republic).

## 3. Results

### 3.1. Visual and Microscopic Evaluation

Visual evaluation of bee pollen loads confirmed significant variability in color for the samples assigned to sunflower, goat, willow, and opium popper that could be proof of their heterogeneity. By contrast, bee pollen loads assigned to blue tansy were uniform in color. SEM images of bee pollen loads may confirm or decline the assumption about its homogeneity (Figure 1A). Homogeneous loads can be assigned to a specific botanical source based on the morphology of pollen grains [9,10]. Pollen admixtures in bee pollen loads could be easily detected by SEM (Figure 1B). Another approach is the releasing of pollen grains by suspending of bee pollen loads in methanol. The isolated grains could be easily identified by light microscopy [11]. Electron and light microscopies were effectively applied for testing of selected bee pollen loads representative for the whole material or its fractions.

### 3.2. Vibrational Spectroscopy

Average FT-MIR ATR spectra of bee pollen samples (loads) are shown in Figure 2A. Arrows identify characteristic bands of the main bee pollen constituents—proteins, fats, organic acids, polyphenols, and carbohydrates [8,9,15,16]. Spectral differences represent no similarities in composition and ratio between the main constituents, i.e., proteins, sugars, fats, aromatics, etc. The spectra were normalized in the region of carbohydrate absorption (1200–950 cm^−1^, CO and CC stretching) that was pronounced for all samples. The spectra have several characteristic bands assigned to the main bee pollen constituents—proteins, fats, organic acids, phenolic compounds, and carbohydrates [8,9,15,16]. Spectral differences between the samples represent non-similarities in composition and ratio between the main constituents, i.e., proteins, sugars, fats, aromatics, etc. The linden bee pollen was rich in proteins (1653 and 1549 cm^−1^, amide I and amide II vibrations). By contrast, loads from white clover contained more fat (2925, 2855, and 1739 cm^−1^, CH_2_ and C=O stretching), loads from opium poppy contained more organic acids (1710 cm^−1^, C=O stretching) [8,16,17,20,21]. Phenolic acids (1516 cm^−1^, C=C stretching) were found in many samples [22]. All spectra demonstrate several bands in the region of 921–780 cm^−1^ assigned to amorphous fructose, the main sugar component of flower nectar [23].

Average FT-NIR DR spectra of bee pollen samples (loads) are shown in Figure 2B. Like in the case of FT-MIR, spectral differences clarify the variability in the chemical composition of the loads. Band assignment in NIR region is very complicated because of the combination and overtone bands overlapping. Arrows identify the characteristic combination and overtone bands of the main bee pollen constituents—proteins, fats, organic acids, phenolic compounds, and carbohydrates [24]. The spectra are sensitive to fat to sugar ratio, probably to protein contribution as well.

Average FT-Raman spectra of bee pollen samples (loads) are shown in Figure 2C. Spectral differences represent non-similarities in composition and ratio between the main constituents and pigments, i.e., proteins, sugars, fats, aromatics, and carotenes. Arrows identify characteristic bands of proteins, fructose, and bee pollen pigments—flavonoids and carotenoids. The carotene bands at 1520, 1155 at 1005 cm^−1^ demonstrate maximal intensity for sunflower and goat willow bee pollen [25]; the last band is overlapped by the protein band near 1004 cm^−1^ (ring breathing of Phe). These bands were intense only for brown loads of pollen from opium poppy, so there is a tight dependence between carotene level and color. Next, intense bands at 1654, 1605 at 1440 cm^−1^ were assigned to amide, aromatic, and aliphatic moieties; the last band was used for normalization of the spectra. Several bands at 1454, 1263, 1075, 916, 868, 821, 707, 630, 519, and 421 cm^−1^ are typical for amorphous fructose [23,26]. Comparing the FT-Raman spectra of bee pollen loads and the corresponding botanical pollen grains represented in Figure 2D, it is evident that these bands of fructose are absent in pollen grains obtained from the same plants, while the characteristic bands of carotenoids are present in both bee pollen and pollen grains.

### 3.3. PCA of Spectroscopic Data

Based on microscopic image evaluation and comparing with the individual FT-MIR ATR spectra, seven fractions of bee pollen loads corresponding to specific botanical species were chosen for PCA of the FT-MIR (1800–710 cm^−1^) and FT-NIR (6011–5382 cm^−1^) data. The spectra of samples **BP7** and **BP8** were excluded from the whole dataset owing to high heterogeneity. First, derivatives were used instead of the initial spectra because the former gives better discrimination than the latter. The 3D score graphs demonstrate the different space location of clusters corresponding to specific botanic species. Discrimination was made based on PCA of 1st derivative FT-MIR and FT-NIR spectra by the combination of PC2, PC3, and PC4 (Figure 3A) or PC1, PC2 and PC3 (Figure 3B), respectively.

Results of PCA analysis of the FT-Raman spectra (300–1900 cm^−1^) of ninety loads, i.e., ten loads independently chosen from each of the nine samples, are demonstrated in 2D and 3D score graphs (Figure 4, left and right). Bee pollen samples were discriminated by a combination of the three first PCs. It is evident that PC1 separated samples according to the presence and the amount of carotenoids in some loads of bee pollen (**BP1**, **BP7**, and **BP8**). The next two components were sensitive to the ratio between flavonoids and proteins. Samples **BP4**, **BP5**, and **BP9**_1 showed intense Raman bands of aromatics at 1603 cm^−1^ (C=C stretching) and 1225 cm^−1^ (C-Ph stretching). By contrast, the spectra of samples **BP2**, **BP3**, and **BP7** have weak bands of aromatics mentioned above but a strong band of proteins near 1650 cm^−1^ (amide I). Finely, the spectra of samples **BP1** and **BP6** have weak bands of both proteins and aromatics. Therefore, PCA of FT-Raman spectra led to the discrimination of bee pollens according to their composition with stress pigments (flavonoids, carotenes) and proteins.

### 3.4. Identification of Pollen by Color and Microscopic Methods

Visual evaluation confirmed the significant variability in color for bee pollen loads, which could be proof of their homogeneity inside the samples. For example, **BP5** contained only blue loads assigned to *Phacelia tanacetifolia*, whereas samples **BP2** and **BP8** are made up of several color types (fractions) of loads. Color fractions of bee pollen loads, their portions, and presumable identification of the botanic source are summarized in Table 2. The discrimination of load fractions for samples **BP2** and **BP8** as well as for the other samples was made based on the CIE L*a*b* color diagram {a*; b*} shown in Figure 5A,B. It is evident that there was no perfect distribution in the case of samples **BP2** and BP8, due to the heterogeneity of the samples. Corresponding images of typical loads are demonstrated for each fraction. SEM images of bee pollen loads were used to confirm or deny the assumption about homogeneity, as well as to identify the botanical source of pollen grains. It is evident that greater image resolution permits to analyze the surface of the load, packaging of pollen grains in the load, the morphology of individual grains, and even the structural features of exine (Figure 1A). Homogeneous loads can be assigned to a specific botanical source based on the morphology of pollen grains [9,10]. In addition, pollen admixtures in bee pollen loads were also detected and identified by SEM (Figure 1B). Commonly, pollens of different families, like Rosacea, Asteracea, Papaveracea, and Apiacea, demonstrate significant differences in shape and surface morphology. By contrast, pollens of the same genus or family are often very similar to each other, like those of dandelion (*Taraxacum officinale*, Asteracea) and chicory (*Cichorium intybus*, Asteracea). In such cases, other pollen features like color and blooming season should be taken into account.

Another approach is the releasing of pollen grains by suspending bee pollen loads in methanol. The isolated grains could be easily identified by light microscopy [11]. SEM images of the grains in these color fractions confirmed their botanical origin. For example, polyfloral **BP1** had the main fraction of yellow loads, **1a** (74%) identified as oilseed rape (*Brassica napus*). Two minor fractions of dark green **1b** (10%) and reddish **1c** (6%) loads were identified by SEM images as Ragged robin (*Lychnis floscuculi*) and horse chestnut (*Aesculus hippocastanum*), respectively (Figure 6). Similarly, polyfloral **BP2** (sunflower) consisted of four colored fractions (Table 2): golden yellow **2a** (64%), lemon yellow **2b** (23%), red brown **2c** (17%) and dark yellow **2d** (6%). Only the major fraction **2a** was found to be made of pure sunflower (*Helianthus annui*) pollen grains, the others were polyfloral mixtures with the prevalence of clover (*Trifolia* sp.), wild carrot (*Daucus carrota*) and bellflowers (*Campanula* sp.) (Figure 7). Two minor red-brown fractions of loads were found in **BP3** and **BP7**; in both cases, the loads contained mainly popper (*Papaver* sp.) pollen grains (Figure 8). Another example—polyfloral **BP8**, had three colored fractions of loads: green **8a** (41%), yellow **8b** (39%) and golden yellow **8c** (20%). It was found that the pollen grains of each fraction have a specific origin, defined as plum (*Prunus domestica*), goat willow (*Salix caprea*), and dandelion (*Taraxacum officinale*), respectively (Figure 9). Finally, three bee pollens, **BP9**_1, **BP9**_2, and **BP9**_3, obtained from different localities (fields) were defined by beekeepers as buckwheat (*Fagopyrum esculentum*). However, these samples demonstrated significant differences in color and pollen morphology (Figure 10). These differences could be explained by the high biodiversity in the frame of species. Electron and light microscopies could be, thus, effectively applied for testing of selected bee pollen loads, representative for the whole material or its fractions.

## 4. Conclusions

In this manuscript, we propose a new approach to bee pollen characterization. Firstly, one hundred randomly selected loads are visually evaluated and divided into fractions by color. This is followed by microscopic and vibrational spectroscopic characterization of the chosen individual loads, typical for previously determined color fractions. As a result, the botanical origin and the composition of the loads are assessed. Visual (preliminary) selection according to color followed by image analysis (CIE L*a*b*), and microscopic (SEM) characterization of bee pollen load according to the morphology of pollen grains, respectively, are necessary for the regular evaluation of bee pollen homogeneity. The analysis of light microscopic images of methanol extracts of bee pollen can also be applied for morphological evaluation and distribution of pollen grains. Most of the bee pollen samples used in this study showed moderate or significant heterogeneity in their composition and botanical origin.

Then, based on vibration spectra and using multivariate statistics, a model is created for the discrimination of load fractions depending on the composition, color, morphology of pollen grains, and botanical origin. The results obtained confirmed that the vibrational spectra of bee pollen loads are suitable for assessing the heterogeneity of bee pollen and identifying fractions originating from other botanical species. The FT-MIR, FT-NIR, and FT-Raman spectra are very sensitive to the composition of bee pollen, and the main chemical components of bee pollen load, i.e., proteins, lipids, carbohydrates, etc., were easily detected by the characteristic vibration bands. Differences in pigment composition of bee pollen samples, including the ratio between carotenoids and flavonoids, were assessed better using FT-Raman spectra. This method is known to be more sensitive to the presence of unsaturated and aromatic compounds. These differences were in agreement with CIE L*a*b* color parameters calculated from image analysis. Obtained results will be a prerequisite for further research routed towards a more complete characterization of bee pollen composition and quality control.

To increase the number of botanical species, palynological databases can be used, including microscopic images of individual pollen grains, as well as bee pollen color atlases used by beekeepers and other specialists in this field. However, the analyst should have experience in this area because some botanical species have pollen grains of very similar morphology. In addition, sometimes it is difficult to distinguish colors visually, and, thus, color parameters should be used for objective pollen identification. Pollen color is usually associated with certain pigments, and FT-Raman spectroscopy helps identify suitable chromophores. In addition, the identity of the most characteristic Raman bands for flower and bee pollens of the same botanical species, except for fructose from nectar, would be a good prerequisite for assessing the origin of bee pollen. Unfortunately, we do not know of any suitable FT-IR/Raman bee pollen library for reference use. Therefore, the creation of such a library may be a task for the future.

## Figures and Tables

**Figure 1 foods-10-01682-f001:**
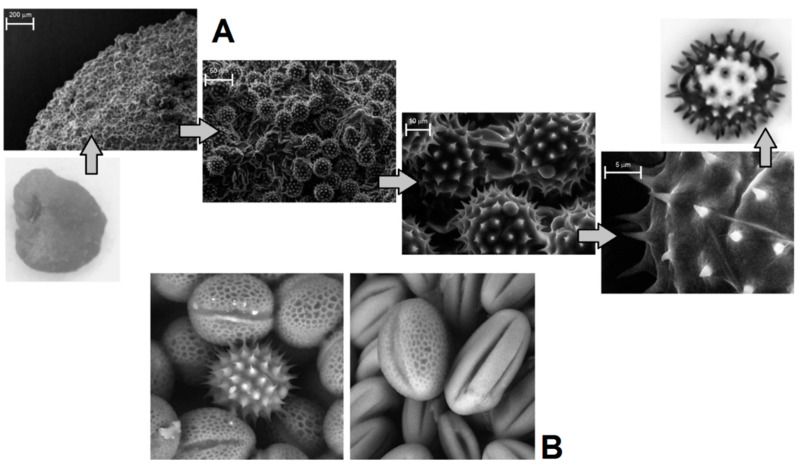
Light microscopic and SEM images of bee pollen: from load to grain (**A**); SEM detection of pollen inhomogeneity (**B**).

**Figure 2 foods-10-01682-f002:**
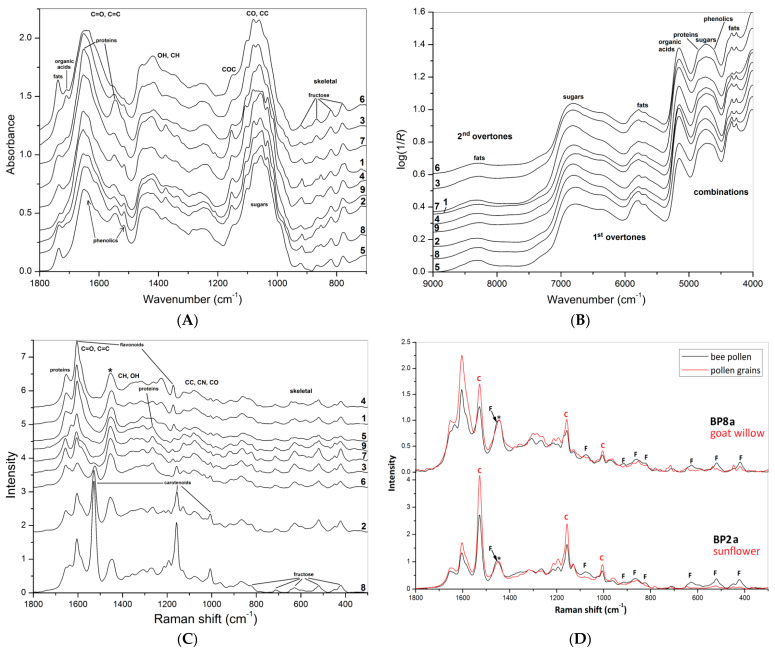
Average FT-MIR (**A**), FT-NIR (**B**), and FT-Raman (**C**) spectra of **BP1**–**8** and **9**_1; comparing FT-Raman spectra (**D**) of bee pollen loads **BP2a**, **BP8a** (black) and corresponding plant pollen grains (red), i.e., sunflower and goat willow, respectively (c—carotenoids, f—fructose), asterisk indicates position of normalized band.

**Figure 3 foods-10-01682-f003:**
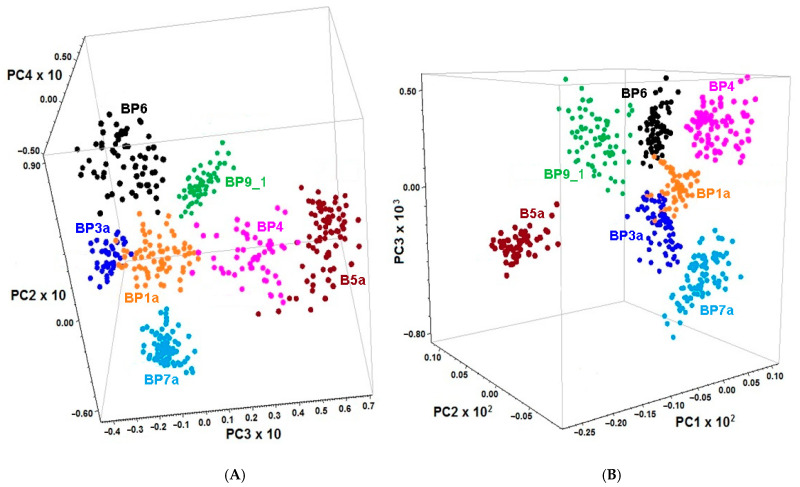
The 3D score graphs for PCA of the 1st derivatives of normalized FT-MIR (**A**) and FT-NIR (**B**) spectra of the loads randomly chosen from the unifloral (**BP4**, **BP6** and **BP9**_1) and the main fractions of polyfloral (**BP1a**, **BP3a**, **BP5a** and **BP7a**) bee pollens.

**Figure 4 foods-10-01682-f004:**
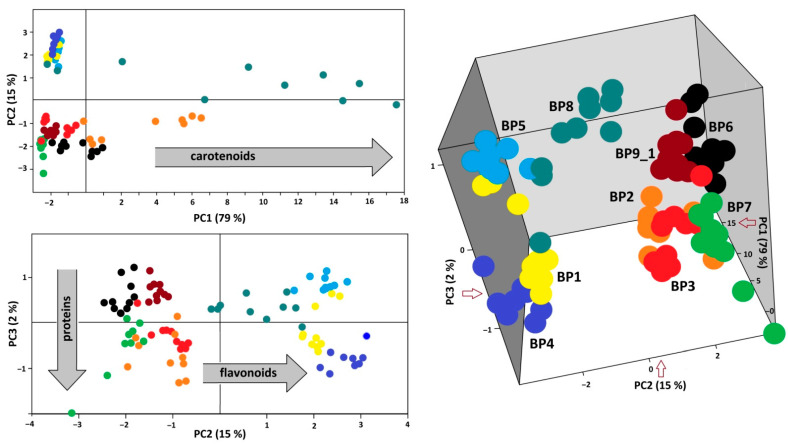
The 2D (left) and 3D (right) score graphs for PCA of FT-Raman data (300–1900 cm^−1^) of **BP1–8** and **BP9**_1.

**Figure 5 foods-10-01682-f005:**
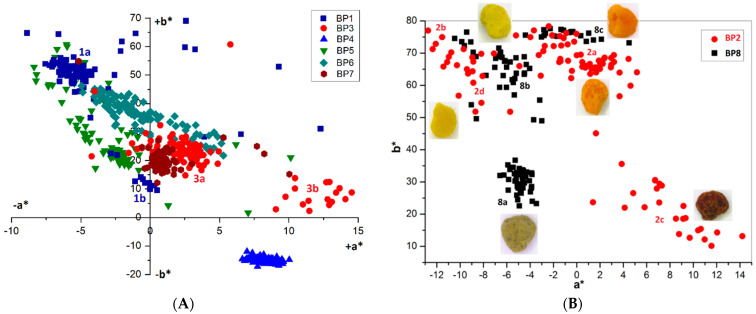
Chromatic diagram {a*; b*} for bee pollen loads: (**A**) **BP1**, **3**–**7** (100 loads per sample); (**B**) **BP2** and **BP8** and images of loads corresponding to the individual fractions.

**Figure 6 foods-10-01682-f006:**
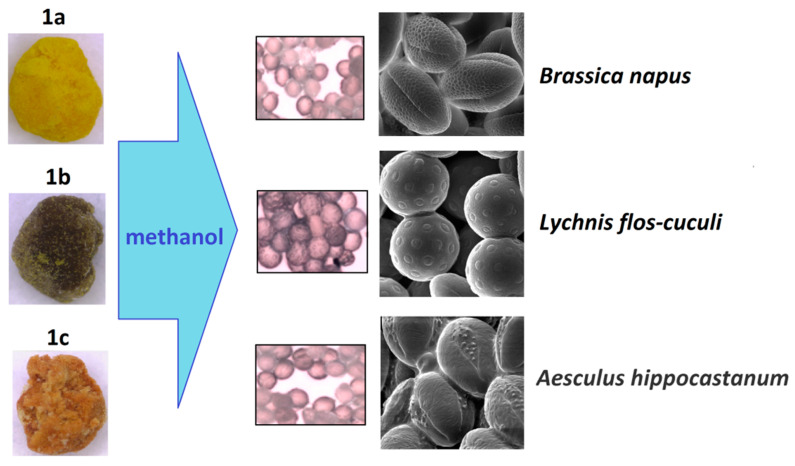
Microscopic identification of pollen grains in methanol extracts from yellow, green, and reddish loads (**1a**–**c**) of **BP1**.

**Figure 7 foods-10-01682-f007:**
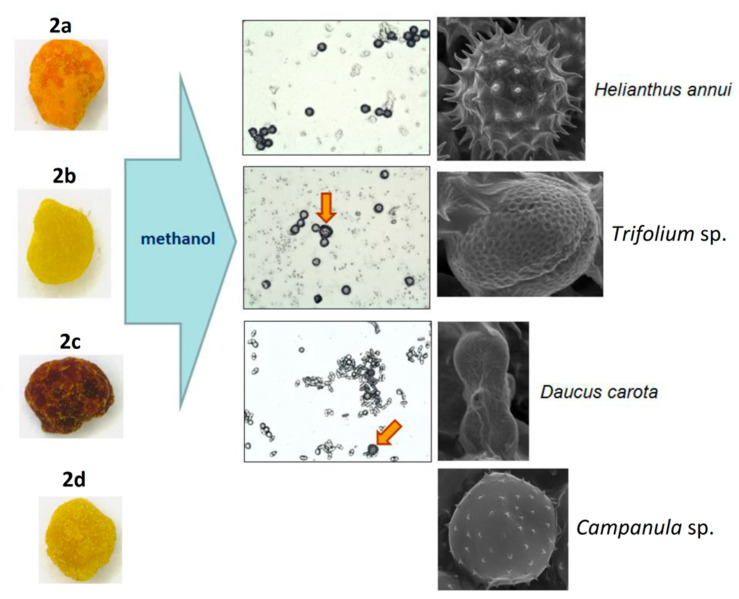
Microscopic identification of pollen grains in methanol extracts from yellow, brown, and orange (**2a**–**d**) loads of **BP2**.

**Figure 8 foods-10-01682-f008:**
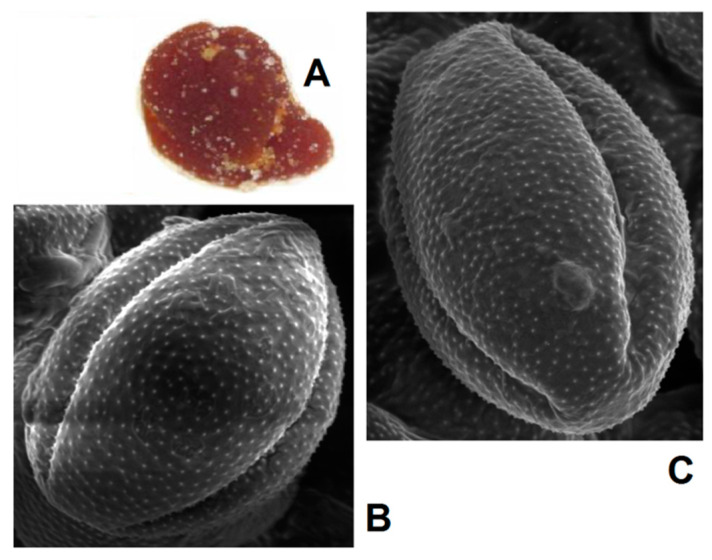
SEM images of pollen grains in brown loads (**A**) found in **BP7** (**B**) and **BP3** (**C**).

**Figure 9 foods-10-01682-f009:**
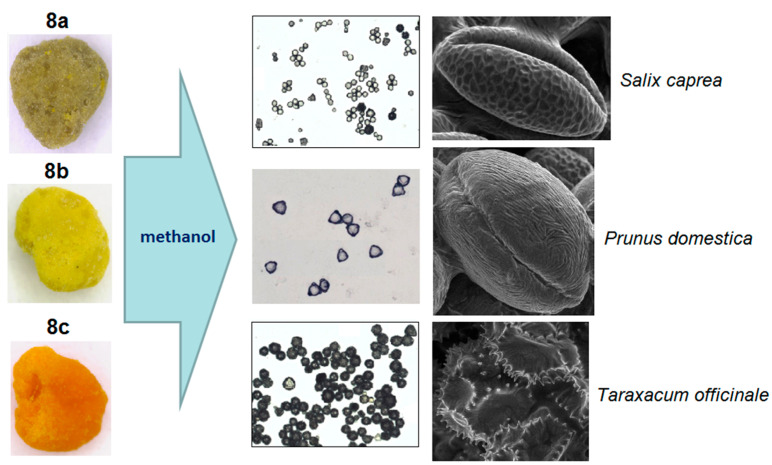
Microscopic identification of pollen grains in methanol extracts from green, yellow and orange loads (**8a**–**c**) of **BP8**.

**Figure 10 foods-10-01682-f010:**
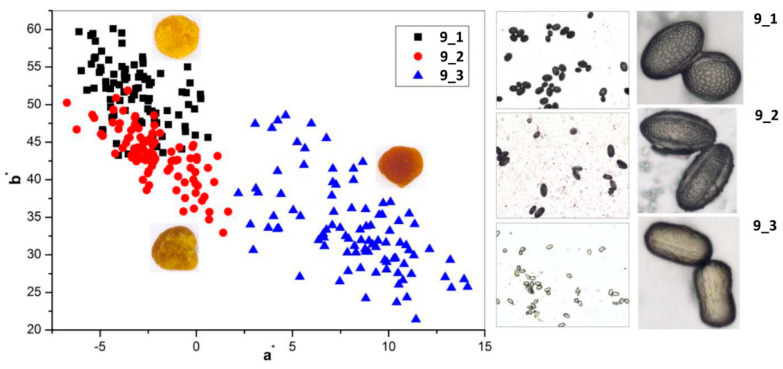
Chromatic diagram {a*; b*} (**left**) and microscopic images (**right**) of unifloral bee pollen loads **BP9**_1, **BP9**_2, and **BP9**_3 (*Fagopyrum esculentum*).

**Table 1 foods-10-01682-t001:** Specification of bee pollen samples.

Sample	Expected Botanical Origin	Harvesting Year	Locality
**BP1**	oilseed rape (*Brasica napus*)	2018	Slepčany (Nitra region, SR)
**BP2**	sunflower (*Helianthus annui*)	2018	Šahy (Nitra region, SR)
**BP3**	opium poppy (*Papaver somniferum*)	2018	Dvory nad Žitavou (Nitra region, SR)
**BP4**	blue tansy (*Phacelia tanacetifolia*)	2018	Zemianske Podhradie (Trenčín region, SR)
**BP5**	black locust (*Robinia pseudoacacia*)	2018	Mošovce (Žilina region, SR)
**BP6**	white clover (*Trifolium repens*)	2018	Tesárské Mlyňany (Nitra region, SR)
**BP7**	linden (*Tilia* sp.)	2018	Nitra city, SR
**BP8**	goat willow (*Salix caprea*)	2018	Hanušovce nad Topľou (Prešov region, SR)
**BP9** (1,2,3)	buckwheat (*Fagopyrum esculentum*)	2018	Šahy (Nitra region, SR)

**Table 2 foods-10-01682-t002:** Fractionation of bee pollen.

Sample	Fraction	Color	Number	Botanical Origin *
**BP1**	**1a**	Yellow	74	*Brassica napus*
	**1b**	dark green	10	*Lychnis flos-cuculi*
	**1c**	golden yellow	6	*Aesculus hippocastanum*
	**1d**	dark yellow	6	ND
	**1e**	Orange	2	ND
	**1f**	light yellow	2	ND
**BP2**	**2a**	golden yellow	64	*Helianthus annui*
	**2b**	lemon yellow	23	*Brassica napus*
	**2c**	red-brown	17	*Daucus carota*
	**2d**	dark yellow	6	*Campanula* sp.
**BP3**	**3a**	yellow-brown	71	*Papaver* sp.
	**3b**	red-brown	16	*Papaver* sp.
	**3c**	Yellowish	3	ND
**BP4**		navy blue	100	*Phacelia tanacetifolia*
**BP5**	**5a**	lemon yellow	96	*Robinia pseudoacacia*
	**5b**	brownish	4	Cichoriacea, *Prunus* sp.
**BP6**		reddish brown	100	*Trifolium repens*
**BP7**	**7a**	brownish yellow	95	*Tilia* sp.
	**7b**	red-brown	4	*Papaver* sp.
	**7c**	lemon yellow	1	ND
**BP8**	**8a**	white green	41	*Salix caprea*
	**8b**	yellow	39	*Prunus domestica*
	**8c**	golden yellow	20	*Taraxacum officinal*
**BP9**_1		yellow	100	*Fagopyrum esculentum*
**BP9**_2		dark yellow	100	*Fagopyrum esculentum*
**BP9**_3		brown	100	*Fagopyrum esculentum*

* ND, not determined

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
