# Peer review of "Spectroscopic Discrimination of Bee Pollen by Composition, Color, and Botanical Origin"

_foods, 2021, doi:10.3390/foods10081682_

Round 1

Reviewer 1 Report

This manuscript reports the identification and discrimination of different types of bee pollen through various spectroscopic methods, including FT-IR, Raman, SEM, etc. Principal component analysis was used to process the IR spectra. In addition, the color test was also applied to identify the different pollens, together with the SEM data. Overall, it is really a good application, and just like the author stated, this characterization is pretty useful for the quality control and authenticity verification. Therefore, this manuscript can be assigned as minor revision.

Here are the questions and concerns.

  1. As it was stated in the abstract “water activity, humidity, and antioxidant capacity of bee pollen was estimated”. Especially for the antioxidant capacity, no such information was observed. Please reconsider it.
  2. It is recommended to list the detail of the bee pollen samples, instead of only listing the origin.
  3. If the bee pollen sample out of these 11 origins is analyzed, how to determination the authenticity? Is there possible to establish or there is an existing bee pollen IR library for the reference use?

Author Response

This manuscript reports the identification and discrimination of different types of bee pollen through various spectroscopic methods, including FT-IR, Raman, SEM, etc. Principal component analysis was used to process the IR spectra. In addition, the color test was also applied to identify the different pollens, together with the SEM data. Overall, it is really good application, and just like the author stated, this characterization is pretty useful for quality control and authenticity verification. Therefore, this manuscript can be assigned as a minor revision.

Here are questions and concerns.

Point 1: As it was stated in the abstract, “water activity, humidity, and antioxidant capacity of bee pollen was estimated”. Especially for the antioxidant capacity, no such information was observed. Please reconsider it.

Response 1: The phrase “water activity, humidity, and antioxidant capacity of bee pollen was estimated” was deleted from the manuscript.

Point 2: It is recommended to list the details of the bee pollen samples, instead of only listing the origin.

Response 2: The information about bee pollen samples of this study was expanded and summarised in Table 1.

Point 3: If the bee pollen sample out of these 11 origins is analyzed, how to determination the authenticity? Is there possible to establish or there is an existing bee pollen IR library for reference use?

Response 3: To increase the number of botanical species, palynological databases can be used, including microscopic images of individual pollen grains, as well as bee pollen color atlases used by beekeepers and other specialists in this field. However, the analyst should have experience in this area, because some botanical species have pollen grains of very similar morphology. In addition, sometimes it is difficult to distinguish colors visually, and thus color parameters should be used for objective pollen identification. Pollen color is usually associated with certain pigments, and FT-Raman spectroscopy helps identify suitable chromophores. In addition, the identity of the most characteristic Raman bands for flower and bee pollens of the same botanical species, with the exception of fructose from nectar, would be a good prerequisite for assessing the origin of bee pollen. Unfortunately, we do not know of any suitable FT-IR / Raman bee pollen library for reference use. Therefore, the creation of such a library may become a task for the future.

Reviewer 2 Report

The manuscript is interesting. The analyses were properly carried out. The procedures used for pollen discrimination showed effectiveness. Some minor revisions are needed.

On Table 1 authors should add the samples’ harvesting years, as well as their geographical origins.

"Results and Discussion" should include a description of the differences or similarities of the samples regarding harvesting years and/or geographical origins . Despite having commented the results regarding composition, colour and botanical origins within “Results and Discussion” paragraph, the conclusions should provide with answers regarding the title and purpose of the manuscript.

Authors should describe the parameters (highlighting the most suitable technique to measure particular features) that help discriminate each bee pollen type by composition, color and botanical origin.

Author Response

Point 1: In Table 1 the authors should add the samples’ harvesting years, as well as their geographical origins. "Results and Discussion" should include a description of the differences or similarities of the samples regarding harvesting years and/or geographical origin.

Response 1: The harvesting years and geographical origins of bee pollen samples were added to Table 1. All samples were collected in 2018 from various localities in Slovak Republic including fields, meadows, gardens, and parks. In this study, unfortunately, there is no space to discriminate bee pollen according to harvesting years because there are no samples harvested in other years for comparison. Similarly, it is impossible to compare samples according to their geographical origin because there are no samples of the same botanical origin but from different localities. This could be done in a new independent study on a specific type of bee pollen, such as sunflower or buckwheat ones, originating from different geographical regions and collected in different years. For this reason, the term "authentication" has been replaced in the manuscript by "identification of botanical origin", which the main task of the presented approach. On the other hand, this is a very interesting research topic that can be used in the future in conjunction with the creation of a special library for authenticating the botanical and geographical origin of bee pollen. Moreover, it can also be used to authenticate honey associated with specific bee pollen. The only thing we were able to do here was to compare the two types of bee pollen (willow and sunflower fractions) with the corresponding pollen grains collected directly from these plants using FT-Raman spectroscopy. The identity of the most characteristic Raman bands for flower and bee pollen of the same botanical species, except for fructose from nectar, would be a good prerequisite for assessing the origin of bee pollen.

Point 2: Despite having commented the results regarding composition, colour and botanical origins within “Results and discussion” paragraph, the conclusions should provide with answers regarding the title and purpose of the manuscript. Authors should describe the parameters (highlighting the most suitable technique to measure particular features) that help discriminate each bee pollen type by composition, color, and botanical origin.

Response 2: The conclusions were expanded and improved according to the reviewer’s comment:

In this manuscript, we propose a new approach to bee pollen characterization. Firstly, one hundred randomly selected loads are visually evaluated and divided into fractions by color. This is followed by the microscopic and vibrational spectroscopic characterization of the chosen individual loads, typical for previously determined color fractions. As a result, the botanical origin and the composition of the loads are assessed. Visual (preliminary) selection according to color followed by image analysis (CIE L*a*b*), and microscopic (SEM) characterization of bee pollen load according to the morphology of pollen grains, respectively, are necessary for the regular evaluation of bee pollen homogeneity. The analysis of light microscopic images of methanol extracts of bee pollen can also be applied for morphological evaluation and distribution of pollen grains. Most of the bee pollen samples used in this study showed moderate or significant heterogeneity in their composition and botanical origin.

Then, based on vibration spectra and using multivariate statistics, a model is created for the discrimination of load fractions depending on the composition, color, morphology of pollen grains, and botanical origin. The results obtained confirmed that the vibrational spectra of bee pollen loads are suitable for assessing the heterogeneity of bee pollen and identifying fractions originating from other botanical species. The FT-MIR, FT-NIR, and FT-Raman spectra are very sensitive to the composition of bee pollen, and the main chemical components of bee pollen load, i.e., proteins, lipids, carbohydrates, etc., were easily detected by the characteristic vibration bands. Differences in pigment composition of bee pollen samples, including the ratio between carotenoids and flavonoids, were assessed better using FT-Raman spectra. This method is known to be more sensitive to the presence of unsaturated and aromatic compounds. These differences were in agreement with CIE L*a*b* color parameters calculated from image analysis. Obtained results will be a prerequisite for further research routed towards a more complete characterization of bee pollen composition and quality control.

To increase the number of botanical species, palynological databases can be used, including microscopic images of individual pollen grains, as well as bee pollen color atlases used by beekeepers and other specialists in this field. However, the analyst should have experience in this area, because some botanical species have pollen grains of very similar morphology. In addition, sometimes it is difficult to distinguish colors visually, and thus color parameters should be used for objective pollen identification. Pollen color is usually associated with certain pigments, and FT-Raman spectroscopy helps identify suitable chromophores. In addition, the identity of the most characteristic Raman bands for flower and bee pollen of the same botanical species, except for fructose from nectar, would be a good prerequisite for assessing the origin of bee pollen. Unfortunately, we do not know of any suitable FT-IR / Raman bee pollen library for reference use. Therefore, the creation of such a library may become a task for the future.
